# Polyphyllin II Induces Protective Autophagy and Apoptosis via Inhibiting PI3K/AKT/mTOR and STAT3 Signaling in Colorectal Cancer Cells

**DOI:** 10.3390/ijms231911890

**Published:** 2022-10-06

**Authors:** Jun-Kui Li, Hai-Tao Sun, Xiao-Li Jiang, Yi-Fei Chen, Zhu Zhang, Ying Wang, Wen-Qing Chen, Zhang Zhang, Stephen Cho Wing Sze, Pei-Li Zhu, Ken Kin Lam Yung

**Affiliations:** 1Department of Biology, Hong Kong Baptist University (HKBU), Hong Kong 999077, China; 2Golden Meditech Center for Neuro-Regeneration Sciences (GMCNS), HKBU, Hong Kong 999077, China; 3School of Traditional Chinese Medicine, Southern Medical University, Guangzhou 510515, China

**Keywords:** polyphyllin II, colorectal cancer, cell cycle arrest, apoptosis, autophagy, PI3K/AKT/mTOR, STAT3

## Abstract

Polyphyllin II (PPII) is a natural steroidal saponin occurring in *Rhizoma Paridis*. It has been demonstrated to exhibit anti-cancer activity against a variety of cancer cells. However, the anti-colorectal cancer (CRC) effects and mechanism of action of PPII are rarely reported. In the present study, we showed that PPII inhibited the proliferation of HCT116 and SW620 cells. Moreover, PPII induced G2/M-phase cell cycle arrest and apoptosis, as well as protective autophagy, in CRC cells. We found that PPII-induced autophagy was associated with the inhibition of PI3K/AKT/mTOR signaling. Western blotting results further revealed that PPII lowered the protein levels of phospho-Src (Tyr416), phospho-JAK2 (Tyr1007/1008), phospho-STAT3 (Tyr705), and STAT3-targeted molecules in CRC cells. The overactivation of STAT3 attenuated the cytotoxicity of PPII against HCT116 cells, indicating the involvement of STAT3 inhibition in the anti-CRC effects of PPII. PPII (0.5 mg/kg or 1 mg/kg, i.p. once every 3 days) suppressed HCT116 tumor growth in nude mice. In alignment with the in vitro results, PPII inhibited proliferation, induced apoptosis, and lowered the protein levels of phospho-STAT3, phospho-AKT, and phospho-mTOR in xenografts. These data suggest that PPII could be a potent therapeutic agent for the treatment of CRC.

## 1. Introduction

Colorectal cancer (CRC) is the third most common cause of cancer mortality worldwide, with more than 1.85 million cases and 850,000 deaths annually [1]. The current therapeutic options for CRC include chemotherapies, targeted therapies, immunotherapies, radiotherapies, and surgery [2]. Among these, surgery is the radical treatment, but it is limited to early treatment. Others such as targeted therapies and chemotherapies have various limitations, such as low response rates, severe side effects, and drug resistance. Thus, the overall 5-year survival rate of CRC is still unacceptably low [2]. Seeking effective and safe therapeutics for CRC treatment is urgently needed.

Cell death has long been associated with cancer therapy. To date, several cell death modalities have been identified and characterized to eradicate cancer cells [3]. Apoptosis and autophagy are two types of programmed cell death. The accumulated evidence has suggested that the pathogenesis of CRC is closed and associated with the signaling pathways involved in regulating apoptosis and autophagy [4]. The signal transducer and activator of transcription 3 (STAT3) is abnormally activated in CRC tumors. It has been reported to regulate the expression of apoptosis-related genes in CRC cells [5]. Therefore, targeting STAT3 signaling has been proposed as an approach for CRC treatment [6]. Autophagy is a key intracellular degradative process. A dysfunction in autophagy closely contributes to the pathogenesis of cancers. A growing number of studies have shown that autophagy exerts paradoxical roles in anti-CRC treatment via multiple signaling pathways, including the inhibition of the PI3K/AKT/mTOR signaling pathway [7]. The PI3K/AKT/mTOR signaling pathway is recognized as a key negative regulatory signal for autophagy [8]. The accumulating evidence suggests that elucidating the role of autophagy in cancer treatment could advance the development of effective interventional strategies for cancer prevention and therapy [9].

*Rhizoma Paridis* is a traditional heat-clearing and detoxifying Chinese herb. Extracts of *Rhizoma Paridis* have been shown to have anti-cancer activities, including against CRC cells [10]. Steroid saponins, such as polyphyllin I (PPI), II (PPII), VI (PPVI), and VII (PPVII), are active ingredients identified in *Rhizoma Paridis*. Among these polyphyllins, PPI has a significant therapeutic effect on hepatocellular carcinoma, CRC, melanoma, prostate cancer, and gastric cancer by inducing cell cycle arrest, apoptosis, and autophagy, as well as by inhibiting the activation of STAT3 and the PI3K/AKT/mTOR signaling pathways [11,12,13,14]. PPVI has been demonstrated to provoke apoptosis and autophagy in human osteosarcoma cells [15]. PPVII is found to retard cell cycle progress and the migration of CRC cells [16]. PPII has been shown to retard the migration and invasion of liver cancer [17]. However, the effects of PPII on CRC are not reported. The underlying mechanism of action also remains elusive. The present study aimed to investigate the anti-CRC effects and underlying molecular mechanism of action of PPII in human CRC cell lines. The in vivo anti-CRC effects of PPII were also explored.

## 2. Results

### 2.1. PPII Inhibits the Proliferation of CRC Cells

Firstly, we investigated the cytotoxic effects of PPII on human CRC cells using MTT assays. The results in Figure 1B show that PPII, in a dose- and time-dependent manner, reduced the viabilities of HCT116 and SW620 cells (Figure 1B). To determine the effects of PPII on cell proliferation, colony formation, and EdU, staining assays were performed, and the results are shown in Figure 1C,D. Treatment with 0.5–2 µM of PPII significantly reduced the number of colonies and the percent of EdU-positive cells in HCT116 and SW620 cells.

### 2.2. PPII Induces G2/M-Phase Cell Cycle Arrest in CRC Cells

To determine whether PPII affects cell cycle progression in CRC cells, flow cytometric analyses were conducted. The results in Figure 2A show that PPII (2 µM) exposure resulted in a significant increase in the cell population in the G2/M phase and a marked decrease in the cell population in the G1 and S phases, suggesting that PPII induces cell cycle arrest at the G2/M phase to impede CRC cell proliferation. Next, we examined the effects of PPII on cell cycle regulatory proteins that participate in regulating the G2/M transition. We found that PPII significantly downregulated the protein levels of cyclin B1, cyclin A2, and CDC2 in CRC cells. Moreover, PPII upregulated the protein level of p21 in CRC cells (Figure 2B). Collectively, these results suggest that PPII arrests CRC cells at the G2/M phase.

### 2.3. PPII Provokes Apoptosis in CRC Cells

A 24 h incubation period with PPII resulted in an increase in apoptotic cells. The percentages of the apoptotic cells increased from 4.7% to 35.2% in the control group and the 4 µM PPII-treated group for the HCT116 cells and from 4.6% to 27.3% in the control group and the 4 µM PPII-treated group for the SW620 cells (Figure 3A). PARP cleavage/activation is a marker of apoptosis. The Western blotting results demonstrated that PPII markedly elevated the protein levels of cleaved-PARP and cleaved-caspase3 in CRC cells (Figure 3B).

### 2.4. PPII Induces Cytoprotective Autophagy in CRC Cells

LC3B is a specific marker of autophagosome formation in mammalian cells, and it is widely used in autophagy measurements [18]. The Western blotting results revealed that the levels of the autophagy-related protein LC3B-II were increased in the PPII-treated HCT116 and SW620 cells (Figure 4A). Next, we used the autophagy inhibitors CQ and Baf-A1 to explore the changes in the LC3B-II protein levels in CRC cells upon PPII treatment. A further increase in LC3B-II protein levels was observed in the CQ− or Baf-A1−plus−PPII−treated cells when compared with the PPII−mono−treated cells (Figure 4B,C). The Western blotting results were confirmed by an immunofluorescent analysis. The LC3 puncta in the PPII-treated SW620 cells were evidently higher than those in the vehicle control-treated cells. In comparison to PPII treatment alone, CQ in combination with PPII significantly increased the number of LC3 puncta in the SW620 cells (Figure 4D). Previous studies have suggested that if autophagy is initiated, co-treatment with CQ will further increase the protein level of LC3B-II; if autophagy is blocked at the late stage, the protein level of LC3B-II will not be affected in the presence of CQ [19]. In line with previous studies, our results demonstrated that PPII activates autophagic flux in CRC cells. The cumulative evidence suggests that autophagy may result in different forms of effects, such as cytoprotective or cytotoxic effects, in response to chemotherapy. Compared with PPII treatment alone, the MTT results showed that the combination of CQ or Baf-A1 with PPII enhanced the inhibitory effect of PPII on cell viability (Figure 4E), suggesting that PPII treatment induces cytoprotective autophagy in CRC cells.

### 2.5. Inhibition of the PI3K/AKT/mTOR Signaling Pathway Is Required for PPII-Mediated Autophagy Initiation in CRC Cells

The inhibition of PI3K/AKT/mTOR activation has been reported to suppress CRC growth, and it is a negative regulator of autophagy. To ascertain whether the PI3K/AKT/mTOR signaling pathway has an important role in PPII−induced autophagy, the protein levels of molecules in this pathway were examined in the PPII−treated CRC cells using Western blotting. As noted in Figure 5A, PPII evidently lowered the protein levels of phospho-PI3K, phospho-AKT (Ser473), and phospho-mTOR (Ser2448) in the HCT116 and SW620 cells when compared with the untreated control. PPII treatment did not significantly change the total PI3K, AKT, and mTOR expression in CRC cells. To further investigate the role of PI3K/AKT/mTOR signaling inhibition in PPII-induced autophagy, rapamycin (rapa, an mTOR inhibitor) was employed. We found that rapa further increased the LC3B−II protein level in PPII−treated cells (Figure 5B). These findings illustrate that the inhibition of PI3K/AKT/mTOR signaling is required for PPII−induced autophagy in CRC cells.

### 2.6. The Inhibition of STAT3 Signaling Is Associated with PPII-Induced CRC Cell Death

The inhibition of STAT3 activation has been reported to suppress CRC cell growth [20]. Here, we found that PPII lowered the protein level of phospho-STAT3 (Tyr705) but did not affect the total protein level of STAT3 in CRC cells. In addition, our results showed that PPII inhibits the phosphorylation/activation of Src and JAK2 (the upstream kinases of STAT3) in the HCT116 and SW620 cells (Figure 6A). To determine the role of STAT3 inhibition in the anti-CRC effects of PPII, the HCT116 cells stably expressing STAT3C were used. As shown in Figure 6B, transfection with STAT3C, in contrast to transfection with an empty vector, remarkably elevated the protein levels of total STAT3 and phospho-STAT3 in the HCT116 cells. The MTT results showed that the cytotoxic effects of PPII against the HCT116 cells were diminished by the STAT3 overactivation (Figure 6B). These results suggest that the inhibition of STAT3 contributes to the anti-CRC effects of PPII.

### 2.7. PPII Inhibits HCT116 Tumor Growth in Mice

Owing to the inhibitory effects of PPII on CRC cell proliferation, we next examined whether PPII suppresses tumor growth in vivo. Treatments with PPII (0.5 mg/kg or 1 mg/kg) significantly suppressed HCT116 tumor growth in mice (Figure 7A,B). When compared with the vehicle control-treated group, the average tumor weight for the 0.5 mg/kg and 1 mg/kg PPII-treated groups was decreased by 73% and 83%, respectively (Figure 7A). The tumor volumes in the PPII-treated groups (0.5 mg/kg or 1 mg/kg) were significantly smaller than those in the control group after dosing for 3 days (Figure 7C). The first line treatment for CRC, 5-FU (30 mg/kg), also caused a significant reduction in tumor volume and tumor weight (Figure 7A,C) compared to the control group. To further define the inhibitory effect of PPII on tumor growth in vivo, Ki-67 expression in xenograft tumor tissues was detected. As can be seen in Figure 7D, PPII and 5-FU treatment predominantly reduced Ki-67 expression in tumors, as indicated by the decreased brown nuclear staining. In addition, a TUNEL assay was conducted to investigate tumor apoptosis. In line with the in vitro results, PPII and 5-FU evidently increased the number of TUNEL-positive cells compared to vehicle control (Figure 7D), suggesting that PPII induces apoptosis in tumors. The Western blotting results showed that PPII lowered phospho-AKT, phospho-mTOR, phospho-STAT3, phospho-Src, and Bcl-XL and upregulated the LC3B-II protein level in tumor tissues (Figure 7E).

## 3. Discussion

*Rhizoma Paridis* plays momentous roles in traditional Chinese medicine, and a large number of studies has confirmed that the main anticancer components of *Rhizoma Paridis* are steroidal saponins [21,22]. The steroidal saponins of *Rhizoma Paridis* have been demonstrated to induce cell cycle arrest, apoptotic cell death, and autophagy in human lung cancer cells [23]. We investigated whether PPII could induce cell cycle arrest, apoptosis, and autophagy in CRC cells. We found that most of the cells were arrested at the G2/M phase following PPII treatment. The proportion of apoptotic cells was significantly increased as the concentration of PPII became elevated, indicating the proapoptotic effects of PPII in CRC cells. Furthermore, we found that PPII increased the number of LC3 puncta and the protein levels of LC3B-II in CRC cells. Autophagy induction refers to an increase in autophagic flux rather than the presence of increased autophagy markers. Therefore, the changes in autophagic flux following PPII treatment were detected using autophagy inhibitors [24]. A further increase in the LC3B-II protein level was observed in cells treated with a combination of CQ and PPII when compared with those treated with PPII alone. These results suggest that autophagic flux is enhanced under PPII treatment. The accumulated evidence suggests that autophagy plays a paradoxical role in cancer treatment [25,26]. Experiments using the autophagy inhibitor CQ showed that autophagy inhibition augmented PPII-induced cell death in CRC cells, suggesting that PPII-mediated autophagy promotes CRC cell survival. A combination of PPII with an autophagy inhibitor would enhance the anti-CRC effects of PPII.

PPII has been demonstrated to sensitize gefitinib to lung cancer cells by inhibiting PI3K/AKT/mTOR signaling [27], a canonical negative autophagy regulating signal. In line with this report, we found that PPII decreased the protein levels of p-AKT and p-mTOR in both CRC cells and the HCT116 xenografts. We further showed that PPII-induced autophagy is associated with the inhibition of PI3K/AKT/mTOR signaling, as evidenced by the enhanced reduction of p-mTOR and the increment of LC3B-II in PPII-treated CRC cells in the presence of an mTOR inhibitor rapamycin compared to those treated with PPII alone. In addition to PI3K/AKT/mTOR signaling, other signaling pathways are also involved in the induction of autophagy, for example, the anti-apoptotic protein Bcl-2. Bcl-2 has been widely reported to interact with Beclin-1, a signaling hub in the context of autophagy, to modulate autophagy [28]. In the future, immunoprecipitation will be performed to assess the effects of PPII on the interaction between Beclin-1 and Bcl-2.

STAT3 has been proposed as a pathogenic factor and therapeutic target of CRC [20]. PPI, another important active component extracted from *Rhizoma Paridis*, has been shown to inhibit STAT3 activation in gastric cancer cells [12]. In this study, we found that PPII also suppressed the phosphorylation/activation of STAT3 in human CRC cells and the HCT116 xenografts. The activation of STAT3 at the tyrosine 705 residue can be mediated by the JAK and Src family kinases [29]. As expected, PPII inhibited the phosphorylation/activation of JAK2 and Src in CRC cells. These findings suggest that PPII exerts anti-CRC effects and suppresses STAT3 signaling in CRC cells and xenografts. In the PPII-treated HCT116 cells, STAT3 overactivation increased cell viability, suggesting that STAT3 is a critical target for the antiproliferative effect of PPII in CRC cells. As a transcriptional factor, the activation of STAT3 at the tyrosine 705 residue is essential for the translocation of STAT3 into the nucleus, where it modulates the transcription of target genes, including apoptosis-related genes such as Mcl-1, Bcl-xl, and Bcl-2, and cell cycle-related genes such as p21 and cyclin B1 [30,31]. In this study, PPII lowered the Mcl-1 and Bcl-xl protein levels and evoked apoptosis in CRC cells. Moreover, PPII increased p21, decreased the protein levels of cyclin B1, cyclin A2, and CDC2, and arrested the CRC cell cycle at the G2/M phase. These results indicate that STAT3 inhibition mediated by PPII is involved in anti-tumor effects in CRC cells.

Wang and his colleagues showed that PPII inhibits the proliferation of, and induces apoptosis in, HepGR and HL-7702 cells, suggesting the hepatotoxic potential of PPII [32]. In the present study, we found that PPII (0.5 mg/kg and 1 mg/kg) treatments potently inhibited HCT116 tumor growth in nude mice. In comparison to the vehicle control-treated mice, PPII at 0.5 and 1 mg/kg significantly reduced the body weight of mice after dosing for 16 days (Appendix A), indicating the potential toxicity of PPII at 0.5 and 1 mg/kg in mice. For future clinical administration, the safety of PPII should be further assessed. Furthermore, new medicament research for the targeted delivery of PPII should be investigated [33].

## 4. Materials and Methods

### 4.1. Reagents and Chemicals

The primary antibodies against LC3B, Src, phospho-Src (Tyr416), JAK2, phospho-JAK2 (Tyr1007/1008), STAT3, phospho-STAT3 (Tyr705), PI3K, phospho-PI3K, AKT, phospho-AKT, mTOR, phospho-mTOR, p53, p21, CDC2, cyclin B1, cyclin A2, Mcl-1, Bcl-XL, poly (ADP-Ribose) polymerase (PARP), and β-actin were purchased from Cell Signaling Technology (Beverly, MA, USA). Dimethyl sulfoxide (DMSO), 3-(4,5-Dimethylthiazol-2-yl)-2,5-Diphenyltetrazolium bromide (MTT), and 5-fluorouracil (5-FU) were purchased from Sigma Company (St. Louis, MO, USA). PPII (purity of ≥ 98%, as determined by HPLC) was bought from Chengdu Must Bio-Technology Co. Ltd. (Chengdu, China). The chemical structure of the PPII is shown in Figure 1A.

### 4.2. Cell Culture

Human CRC cell lines (HCT116 and SW620) were obtained from the American Type Culture Collection (ATCC, Manassas, VA, USA). The CRC cells were cultured in Dulbecco’s Modified Eagle Medium (DMEM) containing 10% fetal bovine serum (FBS) and 1% penicillin/streptomycin (Gibco, Grand Island, NY, USA). The cells were grown in a 5% CO_2_ incubator at 37 °C.

### 4.3. Cell Viability Assay

The cytotoxic effects of PPII on CRC cells were determined using MTT assays. After being plated in 96-well plates (4 × 10^4^ cells per well), the CRC cells were treated with various concentrations of PPII (0.5, 1, 2, and 4 μM) for 24 or 48 h and then incubated with MTT solution (5 mg/mL) for 2 h. The formed formazan crystals in each well were dissolved in 100 μL of DMSO. The optical density (OD) was measured at 570 nm using a microplate spectrophotometer (BD Biosciences, San Jose, CA, USA).

### 4.4. Cell Proliferation Assays

Cellular proliferation was determined by clone formation and 5′-Ethynyl-2′-deoxyuridine (EdU) incorporation assays as described previously [34]. For the colony formation assay, 800 cells were plated in 6-well plates and then incubated with different concentrations of PPII (0.5, 1, and 2 μM) for 24 h. Thereafter, the cells were cultured with fresh medium for ~12 days before being stained with 0.3% crystal violet. The colonies were counted and then captured via a digital scanner. A Cell-light^TM^ EdU Apollo^®^576 EdU In Vitro Imaging Kit (Ribobio, Guangzhou, China) was also employed to assess the proliferative capability of the CRC cells, following the manufacturer’s instructions. CRC cells seeded in 4-well plates were exposed to PPII for 24 h. The cells were then incubated with 10 μM EdU for 2 h. The nuclei of the cells were stained with Hoechst33342 for 30 min. The fluorescence images were captured under an inverted fluorescent microscope (Nikon, Japan).

### 4.5. Cell Cycle Analysis

Cells were seeded in six-well plates at a density of 1 × 10^5^ cells/well. After treating with various concentrations of PPII (0.5, 1, and 2 μM) for 24 h, the cells were harvested, washed with PBS, and gently fixed with 70% ice-cold ethanol at −20 °C overnight. The cells were then resuspended in staining buffer (0.5 mL) containing propidium iodide (PI) (25 µL) and RNase A (10 µL) and incubated at room temperature in the dark for 30 min. Afterwards, the cell cycle distribution was analyzed using a flow cytometer (BD Biosciences), with 10,000 events recorded [35].

### 4.6. Apoptosis Assay

The effects of the PPII on apoptotic cell death were examined using an Annexin V/PI apoptosis detection kit (BD Biosciences). The HCT116 and SW620 cells were seeded in 6-well plates and treated with various concentrations of PPII (0.5, 1, 2, and 4 μM) for 24 h. Both the detached and attached cells were harvested, washed twice with PBS, and then resuspended in 100 µL of 1× binding buffer containing 5 µL of FITC annexin V and 5 µL of PI. After incubation in the dark at room temperature for 15 min, 400 μL of 1× binding buffer was added to each tube. The apoptotic cells were then analyzed using a flow cytometer (BD Biosciences), with 10,000 events recorded.

### 4.7. Immunofluorescence Assay

The SW620 cells grown on coverslips in 12-well plates were treated with the indicated drugs for 24 h and then fixed with 4% PFA for 20 min at room temperature. After blocking with 5% BSA containing 0.5% Triton X-100 for 45 min, the cells were incubated with LC3B antibody (1:200) at 4 °C overnight. On the next day, the cells were washed by PBS and incubated with Alexa Fluor 488-conjugated secondary antibody for 1 h at room temperature in the dark. The nuclei were stained with DAPI in antifade medium. The cells were observed under a scanning confocal laser microscope (Leica Microsystems, Mannheim, Germany).

### 4.8. Western Blotting

Proteins in the cells and tumors were extracted as previously described [35]. The proteins were then separated by 10–12% sodium dodecyl sulfate-polyacrylamide gel electrophoresis (SDS-PAGE) and transferred to PVDF membranes. The membranes were blocked with 5% (*w*/*v*) skimmed milk for 1 h and incubated with the corresponding primary antibodies at 4 °C overnight. Subsequently, the membranes were washed with Tris-buffered saline buffer with Tween20 (TBST) and incubated with HRP-conjugated secondary antibodies at room temperature for 1 h. After washing with TBST, the immunoreactive bands were visualized using an enhanced chemiluminescence (ECL) detection kit (Invitrogen, CA, USA), following the manufacturer’s instructions. The grey value of each band was measured using Image J software.

### 4.9. Cell Stable Transfection

The HCT116 cells expressing constitutively active STAT3 (STAT3C) were established as described previously [36]. Briefly, the HCT116 cells were transfected with pcDNA3-STAT3C or empty vector plasmids using Lipofectamine 3000 (Thermo Fisher Scientific, MA, USA) for 48 h. Then, the cells were selected using 1 mg/mL of G418 in DMEM for 14 days to obtain cells stably expressing STAT3C.

### 4.10. Animal Experiments

Eight-week-old male BALB/c-nu/nu mice were purchased from The Chinese University of Hong Kong. All experimental procedures were approved and conducted in accordance with the guidelines of the Committee on the Use of Human and Animal Subjects in Teaching and Research, Hong Kong Baptist University. All mice were maintained in individual ventilated cages in an animal handling room at the Hong Kong Baptist University. A xenograft mouse model was established as described previously [34]. Briefly, each mouse was inoculated subcutaneously in the flank with the HCT116 cells (2 × 10^6^/mice). Seven days after cell injection, the mice were randomly divided into four groups of four and then i.p.-administered with PBS solution containing 5% PEG400 and 5% Tween80 (the vehicle control), PPII (0.5 mg/kg), PPII (1 mg/kg), or 5-FU (30 mg/kg). Amounts of 0.5 or 1 mg/kg of PPII were i.p. administered every 3 days for 12 consecutive days. 5-FU (30 mg/kg), as the positive control, was i.p.-injected every 2 days from day 7 to day 19 after cell implantation, and then the vehicle was i.p.-administered every 3 days for 12 consecutive days. The tumor volume and body weight of each mouse were measured every 3 days. At the end of the experiment, the mice were euthanized by CO_2_ inhalation. The tumors were dissected, weighed, and photographed.

### 4.11. Immunohistochemistry (IHC) Staining and TUNEL Assays

For immunohistochemical staining, slides were deparaffinized, rehydrated, and incubated in 3% H_2_O_2_ to block endogenous peroxidase activity. After antigen retrieval, which was processed by boiling in sodium citrate for 30 min, the slides were blocked using 10% goat serum for 15 min, followed by incubation with specific primary antibodies at 4 °C overnight. The primary tumor samples were immunostained with Ki-67. Then, the slides were washed 3 times, incubated with the second antibody at room temperature for 30 min, washed 3 times, and incubated with diaminobenizidine (DAB) for 3 min. Finally, the nuclei were counterstained with Mayer’s hematoxylin. The apoptotic cells in the tumor tissues were determined using a Fluorometric TUNEL System. The cell nuclei with green fluorescent staining were defined as the TUNEL-positive nuclei.

### 4.12. Statistical Analysis

All data were expressed as means ± SDs. Statistical analysis was determined by one-way ANOVA, followed by Dunnett’s multiple comparisons using GraphPad Prism version 5.0 software (San Diego, CA, USA). A *p*-value of <0.05 was regarded as statistically significant.

## 5. Conclusions

In conclusion, our data demonstrated that PPII exerts anti-CRC effects in cell and mouse models by inhibiting cell proliferation and inducing cell apoptosis. Furthermore, we found that PPII induces cytoprotective autophagy in CRC cells by inhibiting PI3K/AKT/mTOR signaling. The suppression of STAT3 signaling also contributes to the anti-CRC effects of PPII. Additional pharmacological and toxicological experiments are warranted to clarify its potential use in CRC treatment.

## Figures and Tables

**Figure 1 ijms-23-11890-f001:**
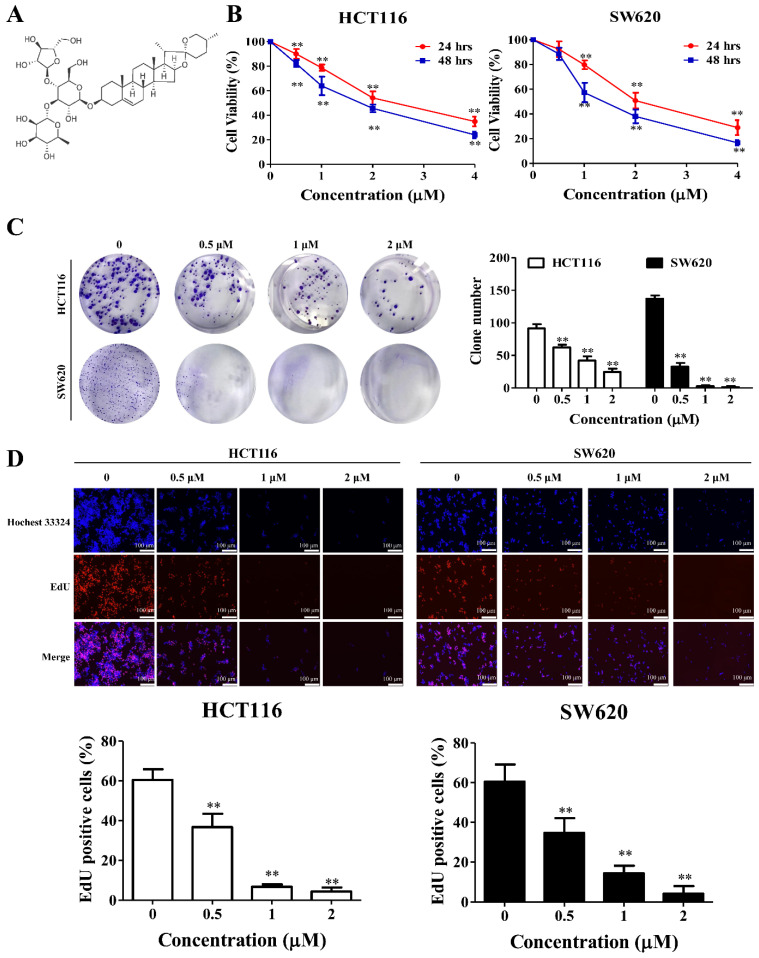
PPII inhibits the proliferation of CRC cells. (**A**) Chemical structure of PPII. (**B**) PPII reduced the viability of CRC cells. Cells were treated with various concentrations of PPII for 24 or 48 h. Cell viability was measured with MTT assays. (**C**) PPII reduced the colonies of CRC cells. The representative results (left panels) and quantitative results (right panel) are shown. (**D**) Effect of PPII on the proliferation of CRC cells was determined using the EdU staining assays. The nuclei were stained using Hoechst33342. The ratio of EdU-positive cells is presented in lower panels. The data shown are the means ± SD of three independent experiments. ** *p* < 0.01 vs. vehicle control.

**Figure 2 ijms-23-11890-f002:**
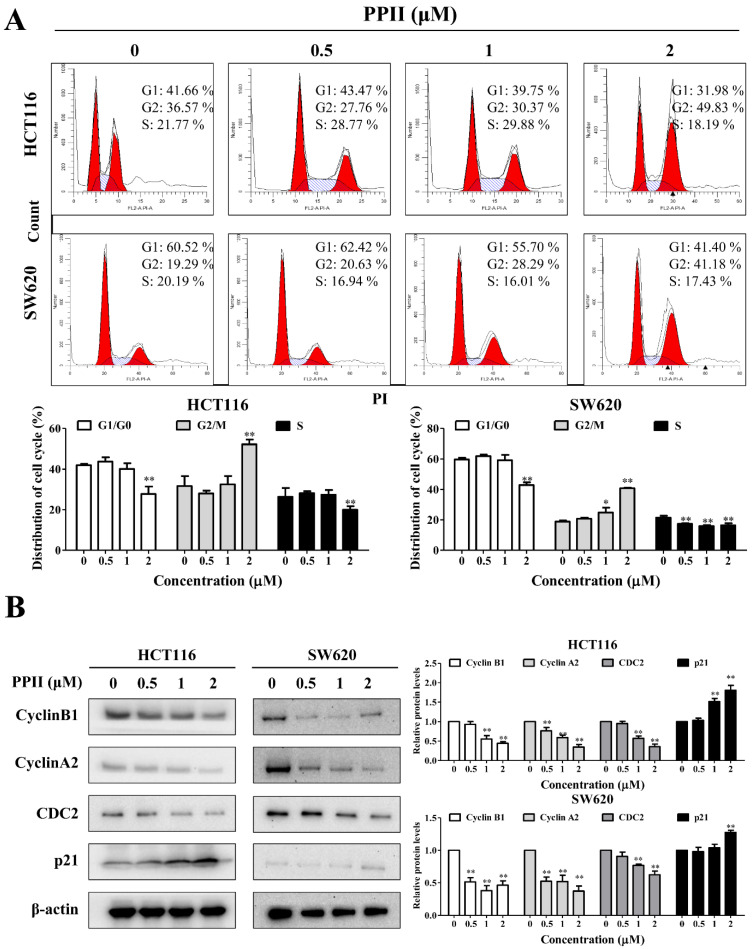
PPII induces G2/M-phase cell cycle arrest in CRC cells. CRC cells were treated with various concentrations of PPII for 24 h. (**A**) PPII-induced G2/M-phase cell cycle arrest in CRC cells. The cell cycle distributions were examined with flow cytometry. (**B**) Effects on cell cycle arrest-related protein levels. The protein levels of cyclin B1, cyclin A2, CDC2, and p21 were determined using immunoblotting. β-actin was included as a loading control. The representative results (left panels) and relative protein levels (right panels) are shown. The data shown in the bar charts are the means ± SD of three independent experiments. * *p* < 0.05 and ** *p* < 0.01 vs. vehicle control.

**Figure 3 ijms-23-11890-f003:**
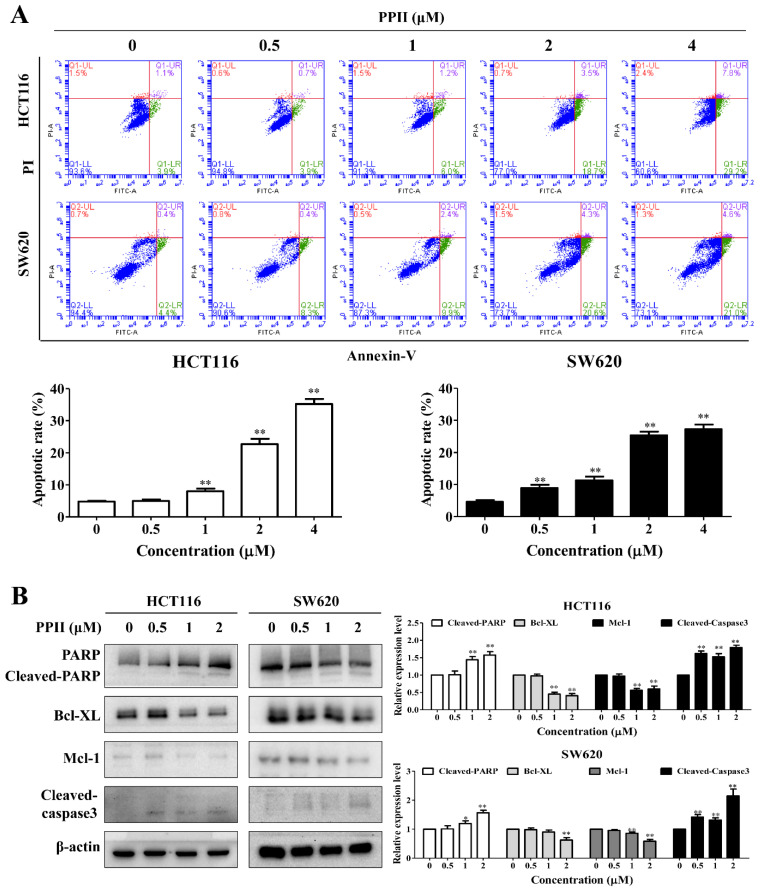
PPII induces apoptosis in CRC cells. The HCT116 and SW620 cells were treated with various concentrations of PPII for 24 h. (**A**) Apoptosis was analyzed using flow cytometry after annexin V/PI double-staining. (**B**) The protein levels of PARP, Mcl-1, Bcl-XL, and cleaved-caspase3 were examined using Western blotting. The left panels show the representative Western blotting results, and the right panels show the quantitative results. The data are presented as the means ± SD of three independent experiments. ** *p* < 0.01 vs. vehicle control.

**Figure 4 ijms-23-11890-f004:**
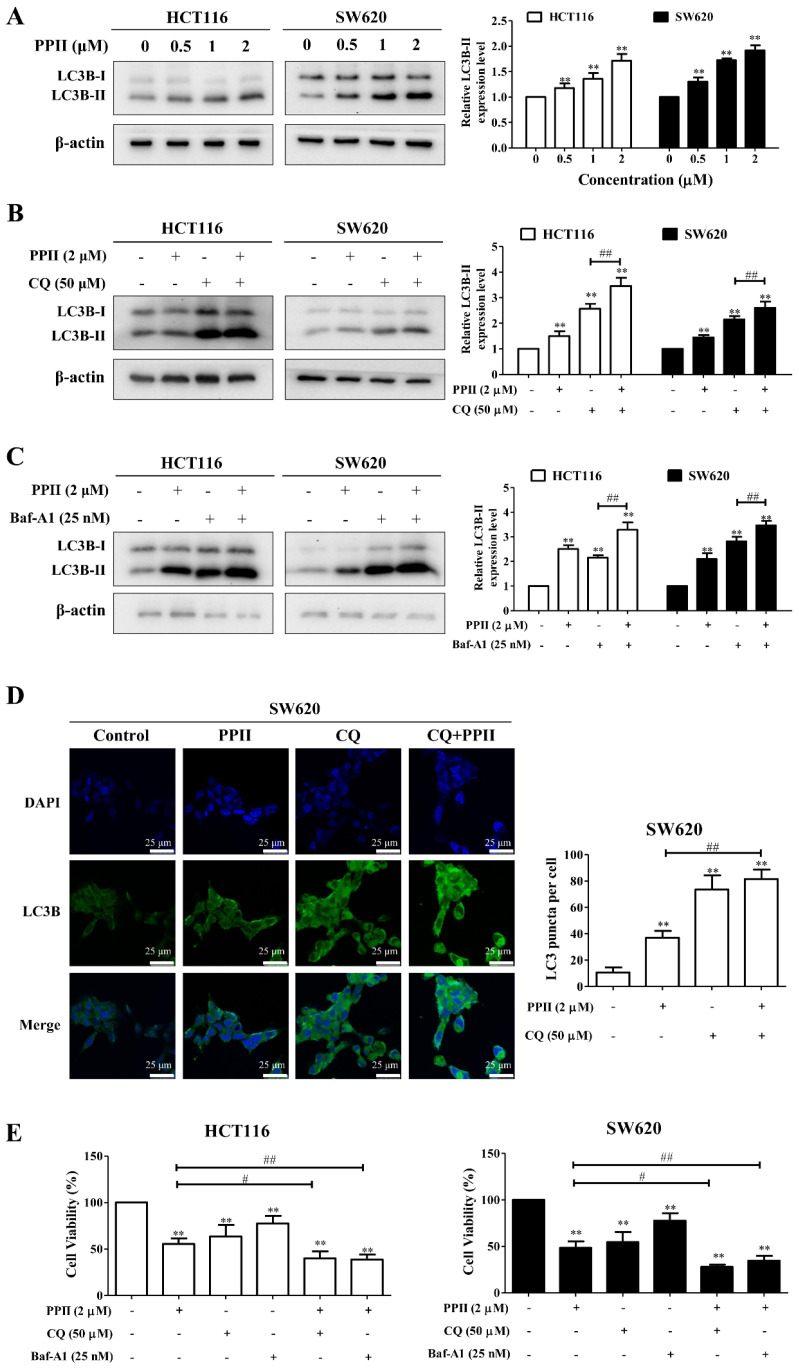
PPII induces autophagy in CRC cells. (**A**) PPII upregulated the protein level of LC3B−II. Representative immunoblotting bands are presented in the left panels, and the quantitative results of the LC3B-II are shown in the right panels. (**B**,**C**) The protein level of LC3B−II in the PPII−plus-CQ− or Baf−A1−treated melanoma cells were examined using Western blotting. (**D**) PPII increased the cytoplasmatic LC3B puncta in the SW620 cells. The cells were treated with or without 2 μM of PPII in the presence or absence of CQ (50 μM) for 24 h. The LC3B puncta were visualized using immunofluorescence analyses. The representative images are photographs taken using a confocal microscope. The scale bar = 10 μm. Representative images (left panels) and the average numbers of green LC3B dots per cell (right panel) are shown. (**E**) The viability of the PPII−plus−-CQ-treated (50 μM) or Baf−A1−treated (25 nM) CRC cells was determined using MTT assays. The data are presented as means ± SD, with *n* = 3. ** *p* < 0.01 vs. the vehicle control; ^#^
*p* < 0.05 and ^##^
*p* < 0.01.

**Figure 5 ijms-23-11890-f005:**
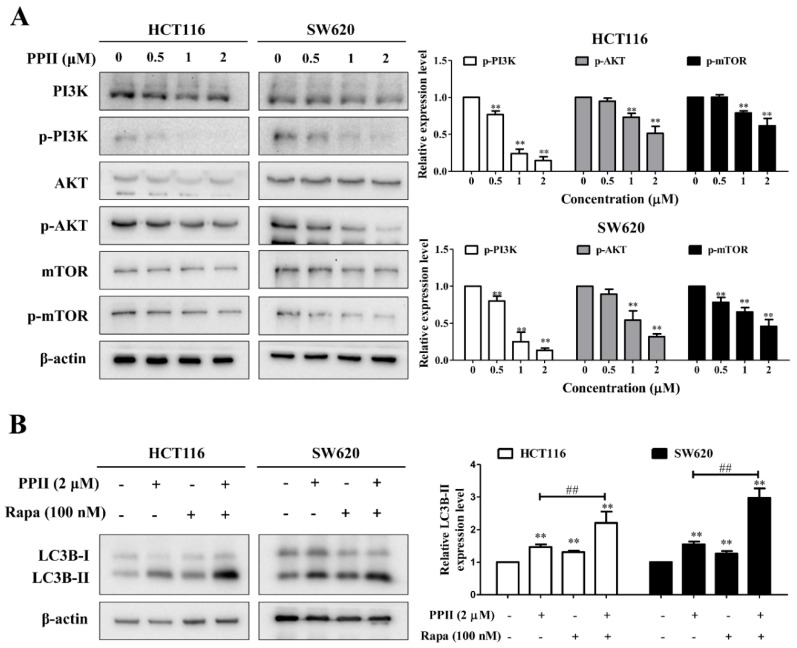
The inhibition of the PI3K/AKT/mTOR signaling pathway is required for PPII-mediated autophagy initiation in CRC cells. (**A**) PPII suppressed the activities of PI3K, AKT, and mTOR in CRC cells. The cells were treated with various concentrations of PPII for 24 h. The protein levels were examined using Western blotting. β-actin was used as a loading control. (**B**) The protein level of LC3B−II in CRC cells exposed to PPII, with or without rapa (100 nm), for 24 h. the representative immunoblotting bands are presented in the left panels, and the quantitative results are shown in the right panels. The data in the bar charts are the means ± SD of three independent experiments. ** *p* < 0.01 vs. vehicle control; ^##^
*p* < 0.01.

**Figure 6 ijms-23-11890-f006:**
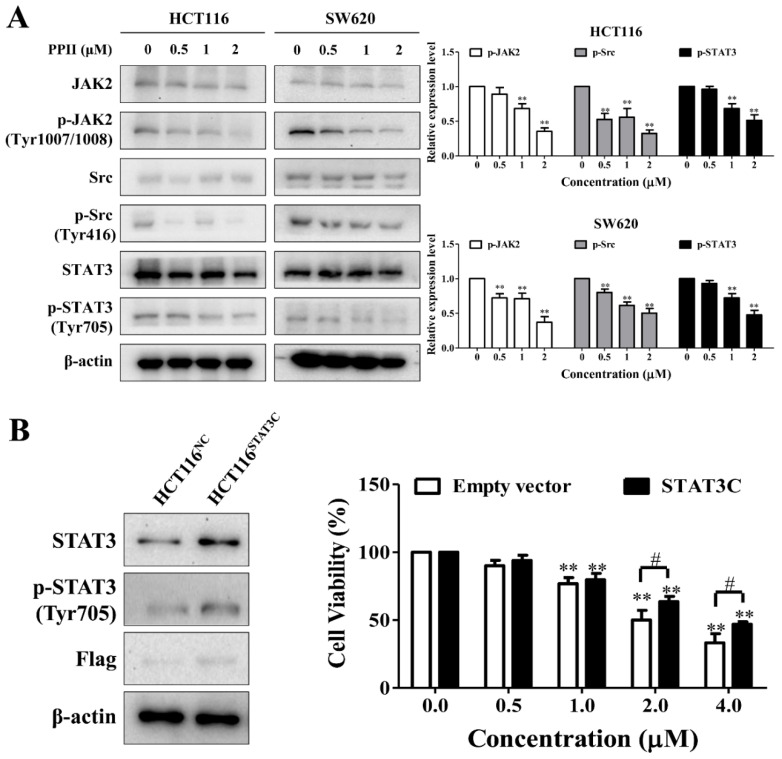
The inhibition of STAT3 signaling is associated with PPII-induced CRC cell death. (**A**) The phosphorylation and total levels of STAT3, JAK2, and Src in CRC cells were examined using Western blotting after exposure to PPII for 24 h. (**B**) The overexpression of constitutively active STAT3 (STAT3C) mitigated the cytotoxicity of PPII against the HCT116 cells. The protein levels and cell viability were determined using Western blotting and MTT assays, respectively. The representative immunoblotting bands are presented in the left panels, and the quantitative results are shown in the right panels. The data in bar the charts are the means ± SD of three independent experiments. ** *p* < 0.01 vs. vehicle control. ^#^
*p* < 0.05.

**Figure 7 ijms-23-11890-f007:**
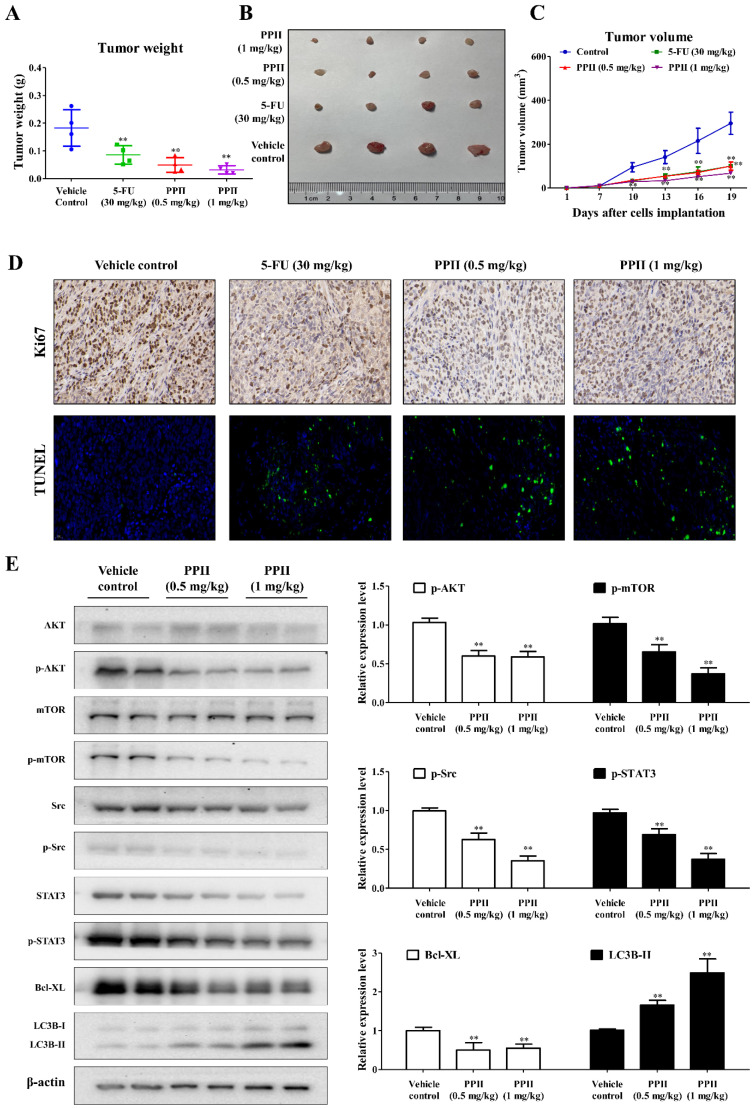
PPII inhibits HCT116 tumor growth in mice. Mice were i.p. administered with the vehicle control (PBS solution containing 5% PEG400 and 5% Tween80), PPII (0.5 mg/kg, once every three days), PPII (1 mg/kg, once every three days), or 5-FU (30 mg/kg, positive control, once every two days) for 12 consecutive days. (**A**) Photographs of the tumors, (**B**) tumor weights, and (**C**) tumor volumes. The data are presented as means ± SDs, with *n* = 4. ** *p* < 0.01 vs. vehicle control group. (**D**) Ki-67 and TUNEL staining of paraffin-embedded tumor sections obtained from the HCT116 xenograft-bearing nude mice. The scale bars = 200 μM. (**E**) PPII decreased the p-STAT3, p-Src, p-AKT, p-mTOR, and Bcl-XL protein levels, but it elevated the LC3B-II protein levels in the HCT116 tumors. The levels of the indicated proteins in the HCT116 allografts from two individual mice in each group were determined using Western blotting. The data shown are means ± SD. ** *p* < 0.01 vs. vehicle control group.

## Data Availability

The data presented in this study are available in article and Appendix A.

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
