# Peer review of "Polyphyllin II Induces Protective Autophagy and Apoptosis via Inhibiting PI3K/AKT/mTOR and STAT3 Signaling in Colorectal Cancer Cells"

_ijms, 2022, doi:10.3390/ijms231911890_

Round 1

Reviewer 1 Report

Jun-Kui Li and colleagues dealt with a topic of particular interest in the context of the possible use of natural compounds in oncological pathologies and in particular the role and effects of P. and colorectal cancer.

The study is interesting, well written and clear. It is supported by appropriate references and  well organized.  

Given the growing interest of Polyphyllin II in oncology and the growing number of studies in the last year (doi: 10.1080/13880209.2022.2120021., doi: 10.21037/tcr-21-966., doi: 10.1039/d1bm01053k., doi: 10.1155/2021/9959634. ), this research offers a good deepening into the initial knowledge (doi: 10.3390/biomedicines10030583., doi: 10.1111/bcpt.13596.) of this natural steroidal saponin in CRC pathogenesis.

In my opinion, this manuscript deserves to be accepted in the present form.

Author Response

We are very grateful to the reviewer for reviewing our manuscript and his affirmation on the manuscript.

Reviewer 2 Report

The authors investigated the natural steroidal saponin polyphyllin II (PPII) as an anti-colorectal cancer (CRC) agent.  Fig. 1 demonstrates PII activity against HCT116 and SW620 CRC cell lines. Fig. 2 shows G2/M cell cycle block and reduction of cyclin B1, cdc2 and increase in p21. Fig. 3 shows data supporting PII induced apoptosis and decrease in anti-apoptotic proteins.  Fig. 4 suggests a PII adaptive/protective increase in autophagy.  Figs. 5 and 6 suggest PII inhibition of PI3K/AKT/mTOR and STAT3 signaling is important in cytotoxic effect. Fig. 7 shows supportive xenograft in vivo effects of PII with corresponding effects found in vitro.

Overall, the manuscript provides useful information on PII in CRC, although there is not much new data compared to previous results using PII in other cancers. The data is thorough, detailed, and well presented.  

A few minor comments:

     1.    Fig. 1C: PII concentrations in histograms do not correspond to images of colonies.

2.    Some figures require better alignment of labeling (Figs. 3B, 4B/C).

3.    Providing some data on toxicity in xenografts such as body weight changes would be helpful.

4.    What is/are the target(s) (if known) for PII?

Author Response

  1. Fig. 1C: PII concentrations in histograms do not correspond to images of colonies.

Response: Sorry for the mistake appeared in the histogram. We amended the concentrations in Figure 1C, and uploaded revised Figure 1. in the revised version.

  1. Some figures require better alignment of labeling (Figs. 3B, 4B/C).

Response: As suggested, we have aligned the labeling in Figures. 3 and 4. The updated Figures were uploaded in the revised version.

  1. Providing some data on toxicity in xenografts such as body weight changes would be helpful.

Response: Thank you for your suggestion. We included the body weight of mice in supplementary materials. And a sentence “In comparison to vehicle control treated mice, PPII at 0.5 and 1 mg/kg significantly reduced body weight of mice after dosing for 16 days (Figure S1), indicating the potential toxicity of PPII at 0.5 and 1 mg/kg in mice. (Figure S1).” was added in the revised version. Please refer to lines 283-285, page 20 and Figure 1S for details.

  1. What is/are the target(s) (if known) for PII?

Response: In the present study, we showed that PPII exerts anti-CRC effects and induces cell cycle arrest, apoptosis and cytoprotective autophagy in CRC cells. The effects of PPII on autophagy is attributed to the inhibition of PI3K/AKT/mTOR signaling. Furthermore, the cytotoxic effects of PPII against HCT116 cells were diminished by STAT3 over-activation, indicating the involvement of STAT3 signaling inhibition in the anti-CRC effects of PPII. Overall, PI3K/AKT/mTOR and STAT3 signaling pathways are the targets of PPII in CRC cells. We included the sentences “we found that PPII induces cytoprotective autophagy in CRC cells by inhibiting PI3K/AKT/mTOR signaling. Suppressing STAT3 signaling also contributes to the anti-CRC effects of PPII.” In “Conclusion” section in the original version. Please refer to lines 397-399, page 25 for details.
